# Joint modelling of systolic and diastolic blood pressure and its associated factors among women in Ghana: Multivariate response multilevel modelling methods

**Justice Moses K. Aheto** [1,2,3]*, **Tracy Gates** [3], **Rahmatu Babah**[1], **Wisdom Takramah**[1]

**1** Department of Biostatistics, School of Public Health, College of Health Sciences, University of Ghana, Accra, Ghana, **2** WorldPop, School of Geography and Environmental Science, University of Southampton, Southampton, United Kingdom, **3** College of Public Health, University of South Florida, Tampa, Florida, United States of America

* justiceaheto@yahoo.com, jmkaheto@ug.edu.gh

**Data Availability Statement:** All the data utilized in this study is freely available upon making official request to MEASURE DHS Team through the

## Abstract

Elevated blood pressure is the leading cause of cardiovascular diseases related mortality and a major contributor to non-communicable diseases globally, especially in sub-Saharan Africa where about 74.7 million people live with hypertension. In Ghana, hypertension is epidemic with prevalence of over 30% and experiencing continuing burden with its associated morbidity and mortality. Using the 2014 Ghana Demographic and Health Survey, we analyzed data on 4744 women aged 15–49 years residing in 3722 households. We employed univariate and multivariate response multilevel linear regression models to analyze predictors of systolic blood pressure (SBP) and diastolic blood pressure (DBP). Geospatial maps were produced to show the regional distribution of hypertension prevalence in Ghana. Stata version 17 and R version 4.2.1 were used to analyze the data. Of the 4744 woman, 337 (7.1%) and 484 (10.2%) were found to be hypertensive on SBP and DBP, respectively. A combined prevalence of 12.3% was found. Older ages 25–34 (OR 2.45, 95%CI: 1.27, 3.63), 35–44 (OR 8.72, 95%CI: 7.43, 10.01), 45–49 (OR 15.85, 95%CI: 14.07, 17.64), being obese (OR 5.10, 95%CI: 3.62, 6.58), and having no education (OR -2.05, 95%CI: -3.40, -0.71) were associated with SBP. For DBP, we found the associated factors to be older ages 25–34 (OR 3.29, 95%CI: 2.50, 4.08), 35–44 (OR 6.78, 95%CI: 5.91, 7.64), 45–49 (OR 10.05, 95%CI: 8.85, 11.25), being obese (OR 4.20, 95%CI: 3.21, 5.19), and having no education (OR -1.23, 95%CI: -2.14, -0.33). Substantial residual household level differences in SBP (15%) and DBP (14%) were observed. We found strong residual correlation of SBP and DBP on individual women (r = 0.73) and household-level (r = 0.81). The geospatial maps showed substantial regional differences in the observed and reported hypertension prevalence. Interventions should be targeted at the identified high-risk groups like older age groups and those who are obese, and the high-risk regions.

website at http://dhsprogram.com/data/available-datasets.cfm. A simple registration is required to freely access and download the data files for use.

**Funding:** The authors received no specific funding for this work.

**Competing interests:** The authors have declared that no competing interests exist.

## Introduction

The leading cause of cardiovascular disease and premature mortality worldwide, hypertension is the "silent killer" affecting more than a billion people, the majority of whom live in low- and middle-income countries (LMICs) [1, 2]. Hypertension occurs when blood pressure is too high, thus exerting too much force on artery walls leading to a reduction in the critical flow of blood and oxygen throughout the body [3]. Hypertension is a significant risk factor for cardiovascular diseases, including stroke, myocardial infarction, heart failure, and atherosclerosis, which account for most of the non-communicable disease deaths globally [1, 2, 4–6]. Blood pressure consists of two measurements; systolic blood pressure which measures the pressure in the blood vessels when the heart contracts or beats while diastolic pressure measures the pressure in the vessels when the heart rests between beats [3]. Hypertension is diagnosed when a person's systolic blood pressure is $\geq$140mmHg, or diastolic blood pressure $\geq$90mmHg [3].

Termed the "silent killer", hypertension is often asymptomatic, resulting in many people being unaware of their condition [1, 3, 7]. Therefore, hypertension diagnosis from blood pressure measurements is critical for people to know their status, receive necessary treatment, and adapt important lifestyle changes. Diagnosis must be supplemented with education and accessibility of treatment. In 2019, it was estimated that less than half of hypertension cases in sub-Saharan Africa were diagnosed, only 22% of men and 45% of women with hypertension were treated, and only 9% of men and 13% of women had their condition under control [4]. These critical gaps in diagnosis and effective treatment must be addressed to reduce the millions of preventable deaths and years of life lost annually.

Because of its global prevalence, detrimental outcomes, and preventability, the reduction of hypertension has been prioritized around the world leading to significant improvements in most countries [2, 3]. However, little improvements have been made in sub-Saharan Africa despite Africa being home to the highest hypertension prevalence rates globally [2–4, 7]. A significant contributor to the limited progress in addressing hypertension in sub-Saharan Africa is the reality of competing interests for limited resources posed by highly visible communicable diseases such as malaria, HIV/AIDS, and tuberculosis [8]. Studies in Ghana, an Anglophone country in sub-Saharan Africa, have found increasing prevalence of hypertension and associated cardiovascular morbidity and mortality despite the initiation of numerous preventative efforts including the National Health Insurance Scheme, national policy for the Prevention and Control of Non-Communicable Diseases, community-based hypertension improvement program, and May Measurement Months [1, 2, 9]. According to a meta-analysis conducted in 2020, the pooled prevalence of hypertension in Ghana is greater than 30% (30.1% for females and 34.0% for males) [8]. Additionally, it is estimated that the diagnosis rate of hypertension in Ghana is between 16.4% and 54.1%, with only 1.7% to 12.7% of hypertension cases under control [10].

Because of the high prevalence rates, and low diagnosis and control rates, it is critical to understand the numerous factors influencing hypertension, and their relationship to each other, in order to enhance prevention efforts by addressing specific concerns in targeted locations. Hypertension is induced through a conglomeration of effects within a complex network of predisposing, enabling, and reinforcing factors. Many studies have found that predisposing factors contributing to hypertension include genetics, sex, race, age, marital status, type and place of residence, weight, pregnancy, health insurance, employment, educational attainment, and lack of understanding of hypertension [1–3, 5, 7, 8, 11–13]. Enabling factors include alcohol consumption, sedentary lifestyle, tobacco usage, diet, and socioeconomic status [1–3, 5, 7, 8, 11]. Factors reinforcing the presence of hypertension in Ghana include the limited availability and accessibility of healthcare, shortcomings of the medical and insurance system, cultural

perceptions, favorable to being overweight, and cost of healthy lifestyles (including diet and physical activity) [1, 2, 5, 8, 11, 12]. Understanding the influence of these demographic and socio-ecological factors on hypertension is critical to addressing them, reducing prevalence, and improving treatment outcomes.

Enhancing the understanding of local risk factors for hypertension is critical to addressing the deadly hypertension epidemic in Ghana. Previous studies have analyzed risk factors of hypertension in Ghana using univariate models [1, 5, 12, 14]; however, no research has yet analyzed these factors using multivariate response multilevel regression modeling approaches to effectively account for the effects of clustering and multilevel interactions. To support targeted and optimal hypertension policy and intervention strategies by policymakers and program managers, there is a need to employ a more complex model that considers maternal, household, and community factors simultaneously, while investigating their interactions across multiple levels.

To this end, we attempt to develop a sophisticated multivariate response multilevel model to jointly analyze systolic blood pressure (SBP) and diastolic blood pressure (DBP) among women in Ghana to understand the complexity of the data to be able to target interventions to improve prevention, diagnosis, and treatment efforts. The multivariate response multilevel modelling is critical in this study because we are interested in modelling both SBP and DBP outcomes measured on same woman nested within a household and we need to draw conclusions about the degree to which the residual correlations depend on these women and their households. We are also interested in quantifying unobserved household level differences in SBP and DBP outcomes, and to investigate specific effect of a given risk factor across these outcomes simultaneously. Also, the multivariate response multilevel modelling approach will permit us to conduct a single test of a joint effect of a risk factor on these two outcomes without multiplying the effect of Type I error which will lead to higher accuracy and reliability in estimates relative to modelling these outcomes separately, especially when these outcomes are at least moderately correlated based on the empirical data [15, 16].

## Materials and methods

### Data source and study population

The data for the current study was extracted from the Ghana demographic and health survey (GDHS) database. The Ghana demographic and health surveys are nationally representative cross-sectional population-based surveys that are carried out every five years by Ghana Statistical Service (GSS) in collaboration with United States Agency for International Development (USAID) and the demographic and health survey (DHS) program. The Ghana Statistical Service has been mandated to implement demographic and health surveys by collecting, analyzing, and providing accurate information on nutrition, malaria, maternal and child health, fertility, gender, and family planning.

The study design used in demographic and health surveys involves a two-stage sampling technique. The enumeration areas (EA) were selected at random using the probability proportional to size technique at the first stage. The second stage involves the selection of households from the EAs using a systematic sampling technique. The three main DHS questionnaires comprise the household questionnaire, women's questionnaire, and man's questionnaire. The sampling frame used in the GDHS was obtained from the 2010 Ghana Population and Housing Census (PHC) which were commissioned by Ghana Statistical Service. The total number of primary sampling unit or enumeration areas (EAs) selected for the 2014 GDHS was 427. A total of 9396 women aged 15–49 years and 4388 men aged 15–59 drawn from 11,385 occupied households were interviewed [17]. The current study extracted biometrics data from the

women's questionnaire where systolic blood pressure and diastolic blood pressure were measured on the women. Detailed description of the methods and procedures utilized in the survey are available elsewhere [17]. In the present study, we extracted data on 4744 women residing in 3727 households.

## Outcome variable

The current study considered systolic blood pressure and diastolic blood pressure as the two outcome variables. These continuous outcome variables are measured in units of millimeters of mercury (mmHg).

## Covariates

The selection of the covariates for the current study was based on the variables used in similar studies and on their availability in the GDHS database [5, 14, 18]. Thus, the current study considered factors such as current age, vegetable consumption, household size, cooking fuel, water source, toilet type, education, wealth, wall type, and obesity.

## Statistical analysis

Data cleaning, recording and validation were done in STATA version 17. We examined any possible presence of outliers in the quantitative variables using boxplot. We also performed checks on the categorical variables to ensure that the categories were mutually exclusive and exhaustive using a simple frequency distribution. Descriptive statistics were performed to describe the characteristics of the study sample.

Previous studies only applied multilevel model to explore any possible individual-level and household-level variations in these outcomes of interest separately [14, 19, 20]. The inferential statistic in the current study focuses on joint modeling of SBP and DBP and their associated factors within a multilevel framework. Clinically, measurement of high blood pressure/hypertension is based on the combination of systolic blood pressure and diastolic blood pressure. Even though we could specify two separate regression models to estimate the correlates of systolic blood pressure and diastolic blood pressure as in previous studies [14, 19, 20], a reviewed literature indicated the computational advantage of joint modeling of outcomes. Our data emanated from a complex survey where individuals were nested in households and given that we are modelling SBP and DBP jointly, we developed and applied the multivariate response multilevel models [15, 16] as the most appropriate statistical method to analyze the data to answer the study objectives. Thus, we jointly model the SBP and DBP of individual women nested within households while simultaneously adjusting for potential risk factors.

## Model description

Let $y_{ij}$, $i = 1,\ldots,N$, $j = 1,\ldots,n_{ij}$, be the response for the $i^{th}$ subject nested in $j^{th}$ household. Let $Y_{1ij}$ represents systolic blood pressure for the $i^{th}$ subject nested in $j^{th}$ household (SBP), $Y_{2ij}$ represents diastolic blood pressure (DBP) for the $i^{th}$ subject nested in $j^{th}$ household. Thus, the equations specified to jointly model $Y_{1ij}$ and $Y_{2ij}$ continuous data are defined by [11, 12]:

$$Y_{1ij} = \beta_1 X_{ij} + h_{1j} + \varepsilon_{1ij} \tag{1}$$

$$Y_{2ij} = \beta_2 X_{ij} + h_{2j} + \varepsilon_{2ij},$$

where $X_{ij}$ denotes the covariate vectors of fixed effects for the outcome variables, $\beta_1$ and $\beta_2$ denote vector of regression coefficients for SBP and DBP respectively, $h_{1j}$ and $h_{2j}$ denotes the

random effects at the household level for SBP and DBP respectively, $\varepsilon_{1ij}$ and $\varepsilon_{2ij}$ represent the error terms or residuals at the individual level for SBP and DBP respectively.

The joint distribution for the random effects that combines both response trajectories is given by:

$$\begin{bmatrix} h_{1j} \\ h_{2j} \end{bmatrix} \sim MVN(0, G) \tag{2}$$

where $G$ denotes the structure of the variance-covariance matrix for the random effect and is defined as:

$$G = \left( \begin{bmatrix} 0 \\ 0 \end{bmatrix}, \begin{bmatrix} \sigma_{h_1}^2 & \sigma_{h_{12}} \\ \sigma_{h_{21}} & \sigma_{h_2}^2 \end{bmatrix} \right) \tag{3}$$

$$\begin{bmatrix} \varepsilon_{1i} \\ \varepsilon_{2i} \end{bmatrix} \sim MVN \left( \begin{bmatrix} 0 \\ 0 \end{bmatrix}, \begin{bmatrix} \sigma_{\varepsilon_1}^2 & \sigma_{\varepsilon_{12}} \\ \sigma_{\varepsilon_{21}} & \sigma_{\varepsilon_2}^2 \end{bmatrix} \right)$$

The residual correlation [15, 16, 21, 22] between systolic blood pressure (SBP) and diastolic blood pressure (DBP) at the household-level ($AE_H$) and women-level ($AE_W$) were estimated using the covariance matrix of the random effects obtained from the multivariate response multilevel regression models.

For the households, we have:

$$AE_H = \frac{cov(h_1, h_2)}{\sqrt{var(h_1)}\sqrt{var(h_2)}} = \frac{\sigma_{h_{12}}}{\sqrt{\sigma_{h_1}^2}\sqrt{\sigma_{h_2}^2}} \tag{4}$$

and for the individual women, we have:

$$AE_W = \frac{cov(w_1, w_2)}{\sqrt{var(w_1)}\sqrt{var(w_2)}} = \frac{\sigma_{w_{12}}}{\sqrt{\sigma_{w_1}^2}\sqrt{\sigma_{w_2}^2}} \tag{5}$$

The intraclass correlation coefficient (ICC) which coincide with the variance partition coefficient (VPC) under the random intercept model provides information on the proportion of total variance in the outcome variable that could be attributable to the household level [21, 23, 24]. Thus, the VPC was estimated for systolic blood pressure (SBP) and diastolic blood pressure (DBP) at the household level using the equations below:

$$VPC(\text{SBP}) = \rho_1 = \frac{\sigma_{h_1}^2}{\sigma_{h_1}^2 + \sigma_{\varepsilon_1}^2} \times 100 \tag{6}$$

$$VPC(\text{DBP}) = \rho_2 = \frac{\sigma_{h_2}^2}{\sigma_{h_2}^2 + \sigma_{\varepsilon_2}^2} \times 100 \tag{7}$$

where $\sigma_{h_1}^2$ and $\sigma_{h_2}^2$ denote the variance of the random effects at the household-level $h_{1j}$ and $h_{2j}$ respectively, $\sigma_{\varepsilon_1}^2$ and $\sigma_{\varepsilon_2}^2$ denote the variance of the residuals at the individual level $\varepsilon_{1ij}$ and $\varepsilon_{2ij}$ respectively.

The joint and separate models were evaluated using Akaike Information criterion (AIC), Bayesian Information criterion (BIC) and likelihood ratio tests (LRT). The model with the lowest AIC and BIC score was selected as the most preferred model. Also, a significant test on

LRT was used to select the preferred model. We examined the residuals for multivariate normality assumption using quantile-quantile plots. The *runmlwin* [25] program that was designed to run MLwiN multilevel models from within Stata was used to fit both the univariate response multilevel and the multivariate response multilevel regression models in Stata version 17.

### Geospatial mapping of the observed and reported hypertension prevalence by regions in Ghana

We downloaded and extracted regional shapefiles for Ghana using Geographic Information Systems (GIS) and geospatial approaches. The shapefiles for the 10 regions that were in existence at the time of the survey were obtained from the GADM website freely available at https://gadm.org/download_country40.html and terms of use/license available at https://gadm.org/license.html. This will allow geospatial mapping of the observed hypertension prevalence based on SBP, DBP, and those ever told to have had hypertension. The packages *rgdal*, *tmap*, and *leaflet* in R version 4.2.1 were used to prepare the data and for the geospatial mapping.

### Ethics approval and consent to participate

The protocol for the 2014 GDHS was reviewed and approved by the Ghana Health Service Ethical Review Committee and the Institutional Review Board of ICF International. An informed consent statement was read to the respondent, who accepted or declined to participate. The study obtained consent to participate from all participants, and a parent or guardian provided consent prior to participation by a child or adolescent [17]. Though this study did not require ethical approval because the GDHS dataset is publicly available and this study was granted permission to use the data by the MEASURE DHS Program, it is worth mentioning that ethical clearance procedures were strictly adhered to before, during and after the GDHS to ensure protection of human rights as required by the US Department of Health and Human Services. Detailed information on demographic and health surveys (DHS) data and ethical standards are publicly available at http://goo.gl/ny8T6X.

## Results

### Summary of background characteristics

We analyzed data on a total of 4744 women residing in 3727 households out of which 7.1% and 10.2% were found to be hypertensive on SBP and DBP, respectively with a combined (i.e., SBP and/or DBP) prevalence of 12.3%, and 6.9% of the women said they were ever told to have had hypertension. Majority were aged 15–24 years (35.6%) while 12.8% of the participants were obese. A total of 2688 (56.7%) had secondary education or higher and majority of them belonged to poorer/poorest households while 3362 (70.9%) reside in households with more than 3 household members. About 14% of them did not eat vegetables in the past 7 days prior to the survey (Table 1).

### Factors associated with SBP and DBP in the univariate response multilevel models

In the univariate models, we found age, obesity, and education to be associated with both SBP and DBP. Type of cooking fuel was only associated with DBP while toilet type was only associated with SBP. Older ages and being obese were associated with increased SBP and DBP while having no education was associated with decreased SBP and DBP. Use of primitive fuel for

**Table 1. Background characteristics of the study participants.**

| Characteristics | Frequency (%) |
| --- | --- |
| SBP | |
| Not hypertensive | 4407 (92.9) |
| Hypertensive | 337 (7.1) |
| DBP | |
| Not hypertensive | 4260 (89.8) |
| Hypertensive | 484 (10.2) |
| Both SBP and/or DBP | |
| Not hypertensive | 4159 (87.7) |
| Hypertensive | 585 (12.3) |
| Ever told to have hypertension | |
| No | 4418 (93.1) |
| Yes | 326 (6.9) |
| Age (years) | |
| 15–24 | 1688 (35.6) |
| 25–34 | 1452 (30.6) |
| 35–44 | 1179 (24.9) |
| 45–49 | 425 (9.0) |
| Obesity | |
| Not obese | 4138 (87.2) |
| Obese | 606 (12.8) |
| Cooking fuel | |
| Advanced | 890 (18.8) |
| Transition | 1439 (30.3) |
| Primitive | 2415 (50.9) |
| Education | |
| Secondary/higher | 2688 (56.7) |
| No education | 1160 (24.4) |
| Primary | 896 (18.9) |
| Wealth | |
| Richer/richest | 1689 (35.6) |
| Poorer/poorest | 2071 (43.7) |
| Middle | 984 (20.7) |
| Water source | |
| Improved | 668 (14.1) |
| Unimproved | 4076 (85.9) |
| Toilet type | |
| Improved | 1707 (36.0) |
| Unimproved | 3037 (64.0) |
| Household size | |
| >3 members | 3362 (70.9) |
| 1–3 members | 1382 (29.1) |
| Wall type | |
| Durable materials | 1431 (30.2) |
| Non-durable materials | 3313 (69.8) |
| Days ate vegetable in the past 7 days | |
| Ate vegetable at least once | 4094 (86.3) |
| Didn't eat vegetable | 650 (13.7) |

cooking was associated with decreased DBP and having unimproved toilet in households was associated with increased SBP. We found substantial residual household level differences in SBP and DBP of 15% and 14%, respectively (Table 2).

**Table 2. Parameter estimates for separate univariate response multilevel models of SBP and DBP.**

| | Parameter estimates | |
|---|---|---|
| | **SBP (95% CI)** | **DBP (95% CI)** |
| **Characteristics** | | |
| Age (years) | | |
| 15–24 | Reference | Reference |
| 25–34 | 2.45 (1.27, 3.63) | 3.28 (2.49, 4.07) |
| 35–44 | 8.72 (7.43, 10.01) | 6.77 (5.91, 7.64) |
| 45–49 | 15.85 (14.06, 17.63) | 10.04 (8.84, 11.24) |
| Obesity | | |
| Not obese | Reference | Reference |
| Obese | 5.09 (3.61, 6.57) | 4.21 (3.22, 5.20) |
| Cooking fuel | | |
| Advanced | Reference | Reference |
| Transition | -0.01 (-1.53, 1.50) | -0.46 (-1.48, 0.56) |
| Primitive | -0.75 (-2.75, 1.26) | -2.06 (-3.40, -0.72) |
| Education | | |
| Secondary/higher | Reference | Reference |
| No education | -2.06 (-3.40, -0.71) | -1.22 (-2.13, -0.32) |
| Primary | 0.60 (-0.68, 1.88) | 0.44 (-0.42, 1.29) |
| Wealth | | |
| Richer/richest | Reference | Reference |
| Poorer/poorest | -0.73 (-2.64, 1.17) | -0.14 (-1.41, 1.14) |
| Middle | -0.11 (-1.64, 1.42) | -0.15 (-1.17, 0.87) |
| Water source | | |
| Improved | Reference | Reference |
| Unimproved | -0.68 (-2.17, 0.80) | -0.38 (-1.38, 0.61) |
| Toilet type | | |
| Improved | Reference | Reference |
| Unimproved | 1.75 (0.56, 2.94) | 0.63 (-0.17, 1.42) |
| Household size | | |
| >3 members | Reference | Reference |
| 1–3 members | 0.78 (-0.31, 1.87) | 0.64 (-0.09, 1.37) |
| Wall type | | |
| Durable materials | Reference | Reference |
| Non-durable materials | -0.01 (-1.27, 1.26) | 0.48 (-0.37, 1.32) |
| Days ate vegetable in the past 7 days | | |
| Ate vegetable at least once | Reference | Reference |
| Didn't eat vegetable | 0.31 (-1.06, 1.67) | 0.34 (-0.57, 1.25) |
| **Residual analysis** | | |
| Household level variance (level 2) | 38.64 (23.32, 53.96) | 16.16 (9.30, 23.02) |
| Women level variance (level 1) | 224.74 (207.97, 241.50) | 101.95 (94.39, 109.52) |
| VPC | 15% | 14% |

## Factors associated with SBP and DBP in the multivariate response multilevel models

To begin with, we estimated the empirical correlation between SBP and DBP, and determined its statistical significance. We found a strong positive and statistically significant correlation of 0.76 (p<0.0001) between SBP and DBP. Furthermore, we fitted a single level (i.e., women level) multivariate response model and compared this to the multivariate response multilevel model (i.e., household and women level). We observed that the multivariate response multilevel model provided a better fit to the data compared to the single level multivariate response model ($\chi^2$ = 36.6, p<0.0001), suggesting substantial unobserved household level variations in SBP and DBP based on the multivariate response multilevel regression model after adjusting for the same set of risk factors considered in both models.

The results from the multivariate response multilevel regression model are presented in Table 3. Generally, the results are similar to those presented in Table 1 except the quantification of the residual correlation between SBP and DBP on individual women and their households and testing the impact of a given risk factors across SBP and DBP which is not possible under the separate model results presented in Table 1. We found that older ages 25–34 (OR 2.45, 95%CI: 1.27, 3.63), 35–44 (OR 8.72, 95%CI: 7.43, 10.01), and 45–49 (OR 15.85, 95%CI: 14.07, 17.64), being obese (OR 5.10, 95%CI: 3.62, 6.58), having no education (OR -2.05, 95% CI: -3.40, -0.71), and unimproved toilet facility (OR 1.75, 95%CI: 0.56, 2.93) were associated with SBP. Furthermore, older ages 25–34 (OR 3.29, 95%CI: 2.50, 4.08), 35–44 (OR 6.78, 95% CI: 5.91, 7.64), 45–49 (OR 10.05, 95%CI: 8.85, 11.25), being obese (OR 4.20, 95%CI: 3.21, 5.19), use of primitive cooking fuel (OR -2.06, 95%CI: -3.40, -0.72), and having no education (OR -1.23, 95%CI: -2.14, -0.33) were associated with DBP (Table 3).

Substantial residual household level differences in SBP and DBP were observed where 15% and 14% of the differences observed in SBP and DBP respectively could be attributable to unobserved household level differences after adjusting for the risk factors considered in our models. Furthermore, the residual correlation on individual women and household-level effects for SBP and DBP were 0.73 and 0.81 respectively, suggesting strong residual correlation of these outcomes on women and their households (Table 4). The residual correlation measures whether after accounting for the risk factors in our model, the factors yet to be identified at the women, household or community levels that predict SBP and DBP are the same. Thus, the higher the residual correlation, the more similar these as-yet unidentified factors that could influence both SBP and DBP will be and vice-versa.

In Table 5, we tested the hypothesis that the effect of a risk factor on SBP is the same as on DBP. Our tests showed that the effects of women's age, and toilet type on SBP were significantly different from that of DBP, but no such differences exist for obesity, cooking fuel and women education.

## Geospatial mapping of observed and reported hypertension prevalence

We found remarkable differences in the prevalence of hypertension among women across the regions of Ghana. Based on SBP, the hypertension prevalence was highest in Greater Accra (15.7%) followed by Ashanti (11.6%) and Eastern (11.6%) regions with Upper West recording the lowest prevalence of 5.3%. For DBP, the prevalence was highest in Greater Accra (17.2%) followed by Western (13.8%) with the lowest prevalence of 5.2% recorded in Upper West region. For the combined prevalence based on both SBP and DBP, Greater Accra recorded the highest (16.8%) followed by Western (12.8%) with Upper West recording the lowest (5.1%). For those who were ever told to have had hypertension, majority of them were from Greater Accra (21.8%) followed by Volta (12.3%) with minority coming from the Upper West region (4.0%) (Figs 1–4).

**Table 3. Parameter estimates for multivariate response multilevel models of SBP and DBP.**

| | Parameter estimates | |
|---|---|---|
| | **SBP (95% CI)** | **DBP (95% CI)** |
| **Characteristics** | | |
| Age (years) | | |
| 15–24 | Reference | Reference |
| 25–34 | 2.45 (1.27, 3.63) | 3.29 (2.50, 4.08) |
| 35–44 | 8.72 (7.43, 10.01) | 6.78 (5.91, 7.64) |
| 45–49 | 15.85 (14.07, 17.64) | 10.05 (8.85, 11.25) |
| Obesity | | |
| Not obese | Reference | Reference |
| Obese | 5.10 (3.62, 6.58) | 4.20 (3.21, 5.19) |
| Cooking fuel | | |
| Advanced | Reference | Reference |
| Transition fuel | -0.02 (-1.54, 1.50) | -0.46 (-1.47, 0.56) |
| Primitive | -0.75 (-2.76, 1.26) | -2.06 (-3.40, -0.72) |
| Education | | |
| Secondary/higher | Reference | Reference |
| No education | -2.05 (-3.40, -0.71) | -1.23 (-2.14, -0.33) |
| Primary | 0.61 (-0.68, 1.89) | 0.43 (-0.43, 1.28) |
| Wealth | | |
| Richer/richest | Reference | Reference |
| Poorer/poorest | -0.73 (-2.64, 1.17) | -0.14 (-1.41, 1.14) |
| Middle | -0.11 (-1.64, 1.42) | -0.14 (-1.17, 0.88) |
| Water source | | |
| Improved | Reference | Reference |
| Unimproved | -0.68 (-2.16, 0.81) | -0.39 (-1.39, 0.60) |
| Toilet type | | |
| Improved | Reference | Reference |
| Unimproved | 1.75 (0.56, 2.93) | 0.61 (-0.18, 1.41) |
| Household size | | |
| >3 members | Reference | Reference |
| 1–3 members | 0.78 (-0.31, 1.87) | 0.65 (-0.08, 1.38) |
| Wall type | | |
| Durable materials | Reference | Reference |
| Non-durable materials | -0.003 (-1.27, 1.26) | 0.47 (-0.37, 1.32) |
| Days ate vegetable in the past 7 days | | |
| Ate vegetable at least once | Reference | Reference |
| Didn't eat vegetable | 0.31 (-1.05, 1.67) | 0.34 (-0.57, 1.25) |

**Table 4. Analysis of variance and residual correlation estimates for the multivariate response multilevel models of SBP and DBP.**

| | Variances | | Intra-household correlation coefficients |
|---|---|---|---|
| | **Child** | **Household** | **Explained variation (%)*** |
| SBP | 224.49 | 38.91 | 15 |
| DBP | 101.75 | 16.37 | 14 |
| | Residual correlation | | |
| Child | 0.73 | | |
| Household | 0.81 | | |

*Ratio of household-level variance to total variance multiplied by 100.

**Table 5. Test for equality of risk factor effect across SBP and DBP.**

| Risk factors | Degree of freedom | Chi-Square value | P-Value |
|---|---|---|---|
| **Characteristics** | | | |
| Age (years) | | | |
| 15–24 (Reference) | | | |
| 25–34 | 1 | 4.20 | 0.040 |
| 35–44 | 1 | 18.84 | <0.0001 |
| 45–49 | 1 | 87.51 | <0.0001 |
| Obesity | | | |
| Not obese (Reference) | | | |
| Obese | 1 | 3.06 | 0.080 |
| Cooking fuel | | | |
| Advanced (Reference) | | | |
| Primitive | 1 | 3.57 | 0.059 |
| Education | | | |
| Secondary/higher (Reference) | | | |
| No education | 1 | 3.09 | 0.079 |
| Toilet type | | | |
| Improved (Reference) | | | |
| Unimproved | 1 | 7.64 | 0.006 |

## Discussion

In this study, we set out to jointly analyze SBP and DBP under multivariate response multilevel modelling methods while simultaneously adjusting for critical risk factors, the first of its kind in this area. Furthermore, we produced geospatial maps to show differences in regional spatial distribution of observed hypertension prevalence based on SBP, DBP, both SBP and DBP (i.e., combined), and reported hypertension. The hypertension prevalence based on SBP and DBP were 7.1% and 10.2% respectively, while those who reported ever been told to have hypertension, and hypertension based on the combined prevalence of SBP and/or DBP were respectively 12.3%, and 6.9%. Substantial regional differences in the hypertension prevalence were found for SBP, DBP, the combined prevalence and the reported. The national hypertension prevalence found in this study ranged from 5.1% in the Upper West region to 16.8% in the Greater Accra region. The prevalence based on SBP ranged from 5.3% in the Upper West region to 15.7% in the Greater Accra region. For DBP, the prevalence ranged from 5.2% in the Upper West region to 17.2% in the Greater Accra region and for those who reported that they were ever told to have had hypertension, the prevalence ranged from 4% in the Upper West region to 21.8% in the Greater Accra region. Thus, Upper West and Greater Accra regions consistently recorded the lowest and the highest hypertension prevalence respectively, on all the four outcome measures. The national prevalence based on SBP and DBP, and a combined (i.e., SBP and/or DBP) prevalence among this group of women aged 15–49 in Ghana was lower than the combined prevalence of 20% reported for women aged 15–49 years in Lesotho [18], 41.4% in Guatemala [26], and 33.8% in Ghana [12].

Of particular interest in this study is the quantification of the residual correlation between SBP and DBP on the individual women and their households in which they reside. This residual correlation measures whether after accounting for the risk factors in our model, the factors yet to be identified to predict SBP and DBP are similar or the same. This is critical because one cannot assume that two different outcomes (SBP and DBP in our case) collected on the same woman at the same time residing in a household are not correlated or similar on the same

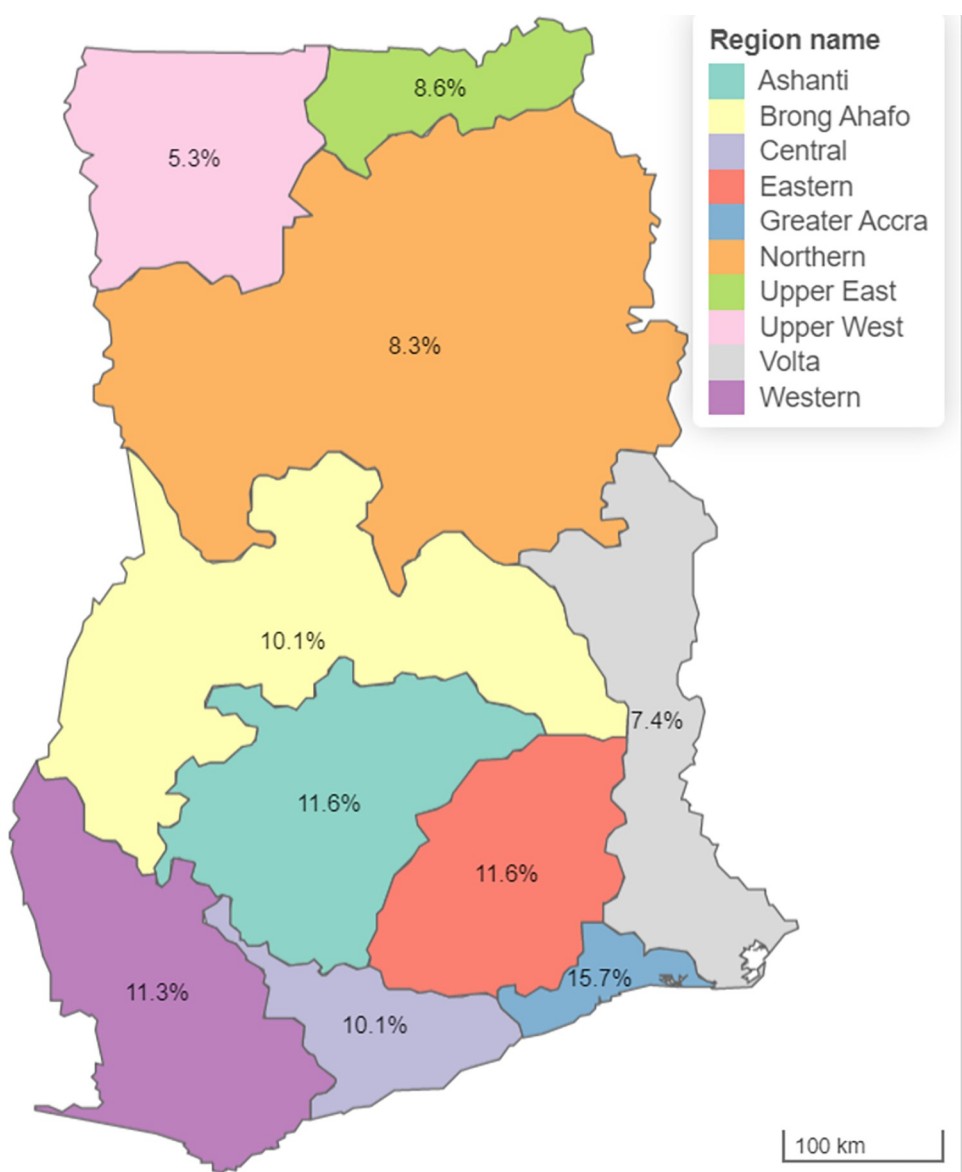

**Fig 1. Regional distribution of observed hypertension prevalence based on systolic blood pressure measurements among women in 2014 in Ghana.** Source: This map was produced by the authors. The link to download the shapefile is available at https://gadm.org/download_country40.html.

woman and her household. Ignoring of this similarity in these outcomes as in the fitting of separate multilevel models for these outcomes could lead to imprecision in model parameters [15, 16, 21]. This is one of the innovations introduced in this study. We found strong residual correlation between SBP and DBP on the individual women and their households, an indication that the as-yet unidentified factors could be predictive of both SBP and DBP. Thus, an intervention being rollout that aimed at identifying other factors to further understand hypertension prevalence based on SBP could similarly address the problem of hypertension based on DBP without the need to rollout separate interventions for SBP and DBP. Also, we are interested in the quantification of the unobserved household level differences in SBP and DBP outcomes which represents variations in household level SBP and DBP outcomes that cannot be

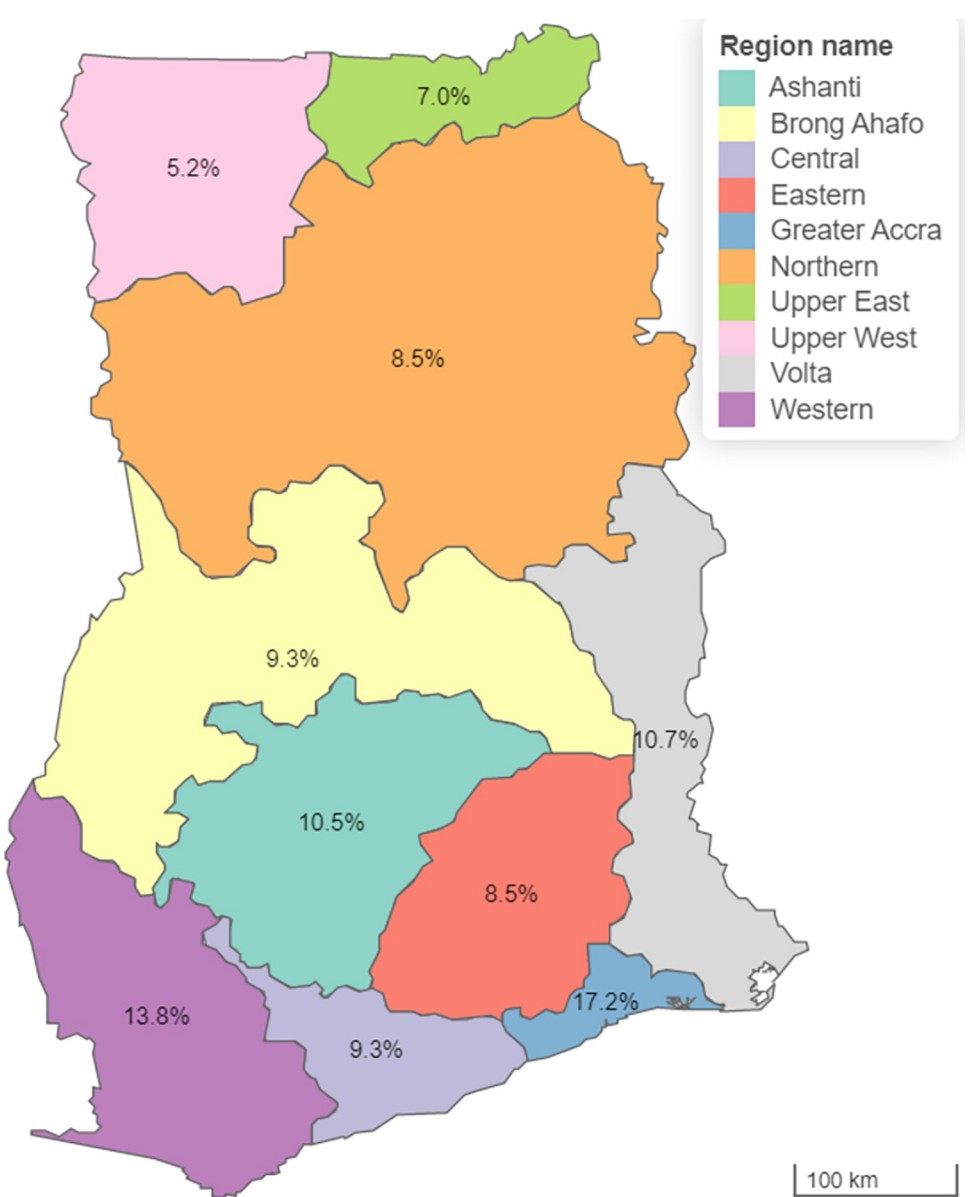

**Fig 2. Regional distribution of observed hypertension prevalence based on diastolic blood pressure measurements among women in 2014 in Ghana.** Source: This map was produced by the authors. The link to download the shapefile is available at https://gadm.org/download_country40.html.

explained by the available risk factors considered in our models. It is a common knowledge that households constitute critical predictors of socioeconomic differences in health and overall well-being of individuals because they influence their opportunities and govern exposure to various opportunities, risks, and resources over their life course [14, 27, 28]. Significant unobserved household-level differences in SBP and DBP among the women were found, suggesting that there are unanswered questions about why a woman residing in a certain household is likely to have increased SBP and DBP compared to a woman residing in another household or vice versa. This finding is consistent with a previous study that found substantial residual household level variations in hypertension prevalence [14].

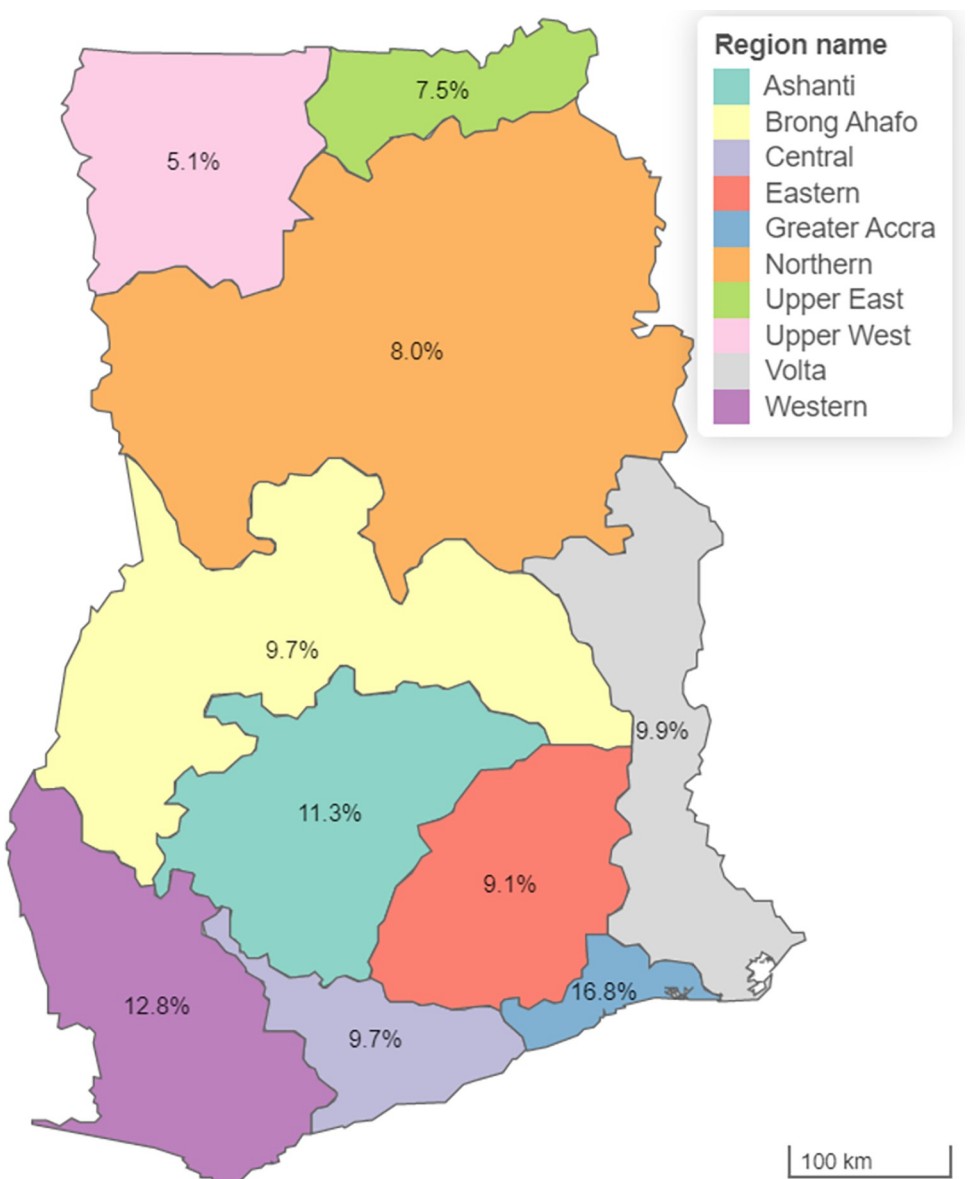

**Fig 3. Regional distribution of observed hypertension prevalence based on both systolic and diastolic blood pressure measurements among women in 2014 in Ghana.** Source: This map was produced by the authors. The link to download the shapefile is available at https://gadm.org/download_country40.html.

Critical risk factors identified to be associated with both SBP and DBP were age, obesity, and education. Specifically, older ages of women and being obese were predictive of increased SBP and DBP, and women with no education had decreased SBP and DBP. Also, having unimproved toilet type in households was associated with increased SBP while using primitive fuel for cooking in households was associated with decreased DBP. Our finding that older ages was associated with increased SBP and DBP is not surprising because generally the older the age, the higher the chance of one developing hypertension and this is consistent with previous studies that found that high blood pressure increase with an increasing age [12, 13, 18, 29–31]. Consistent with previous studies that reported increase in hypertension prevalence among those who are obese [12, 13, 30, 31], being obese was associated with increased SBP and DBP

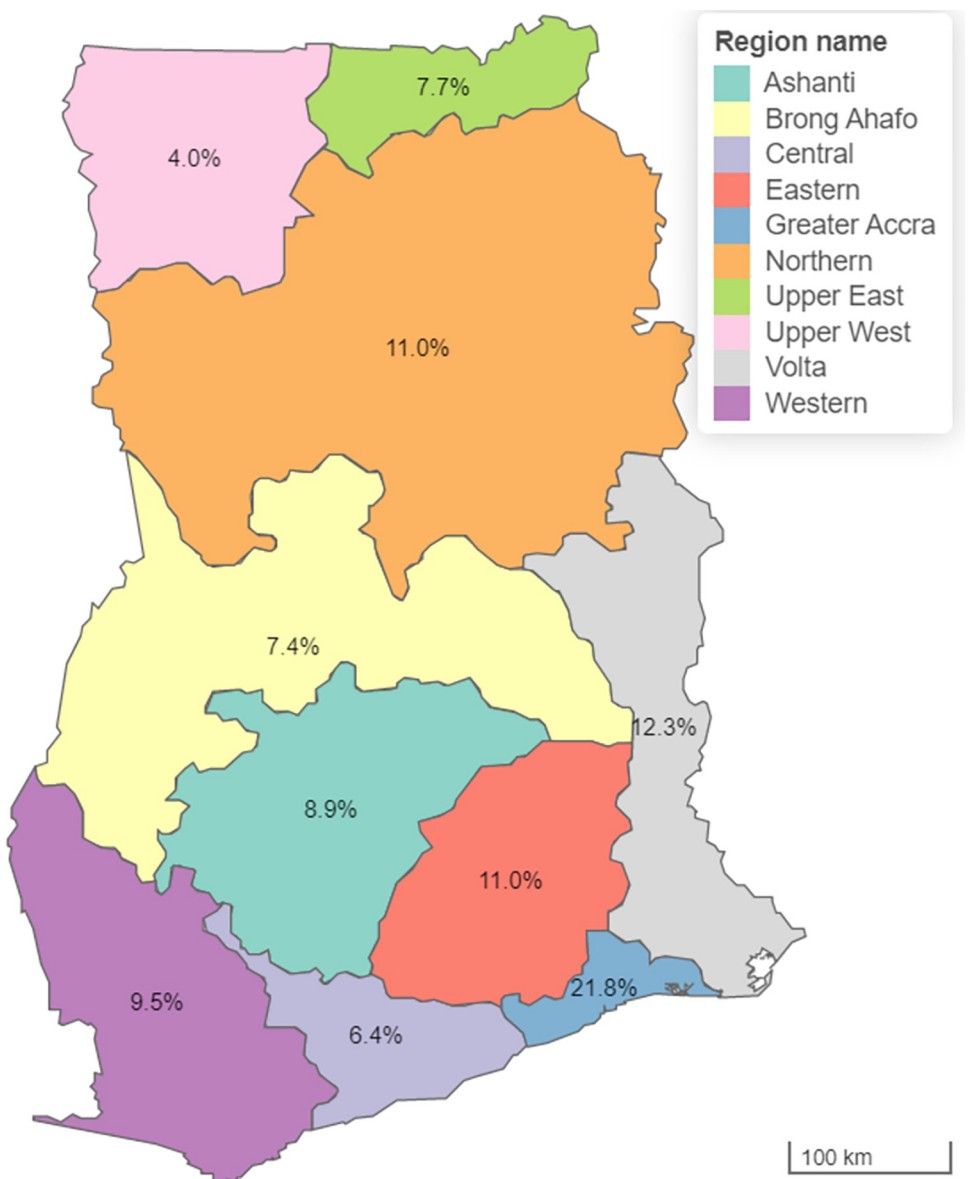

**Fig 4. Regional distribution of reported hypertension prevalence among women who were ever told they had hypertension in Ghana.** Source: This map was produced by the authors. The link to download the shapefile is available at https://gadm.org/download_country40.html.

in this study. Obesity and age were reported to be the most powerful factors associated with hypertension in previous studies [12, 18, 29, 30]. A previous study [31] reported that increase in years of education reduced hypertension prevalence which is not consistent with our finding that women with no education had decreased SBP and DBP. This inconsistency could be attributable to the fact that the study populations are not that similar and that our modelling procedures are more sophisticated with continuous outcomes compared to dichotomous outcome for hypertension used in the previous study [31]. The finding that using primitive fuel for cooking in households was associated with decreased DBP is inconsistent with previous studies which found that biomass fuels for cooking was associated with higher DBP, SBP, and higher risk of hypertension [32, 33]. This could be due to differences in study settings and

statistical modelling approaches. Also, there is lack of available literature on the mechanism through which unimproved toilet facility was associated with increased SBP. Further studies are therefore recommended to identify the possible pathways. The findings also revealed that the effect of women's age, and toilet type differ significantly across SBP and DBP outcomes.

We addressed a critical public health problem facing women of reproductive age, especially in the presence of high maternal mortality in the country as previous studies have reported a link between hypertension versus maternal mortality and morbidity [18]. To improve the general cardiovascular health outcome of women, our modelling approach and the identified risk factors should be considered as part of all overall strategy aimed at addressing hypertension and the general cardiovascular health outcomes in this population. The association of both outcomes (i.e., SBP and DBP) with obesity and type of toilet suggest the need for modification of lifestyle in the form of dietary intake and dietary knowledge and education supported with strong public health nutrition messages, and general improvement in socioeconomic conditions of households. Older women should be targeted with interventions aimed at controlling their high blood pressure while at the same time targeting younger women with interventions aimed at preventing the risk of developing hypertension in their old ages.

## Strengths and limitations of the study

The study is a population-level based with national coverage, making our findings to be readily generalized to the population of women within reproductive age in Ghana and other similar countries. Our modelling approach is very novel and captures residual similarity between SBP and SBP on individual women and their households while simultaneously quantifying unobserved household level differences in these outcomes after adjusting for selected risk factors, leading to more precise estimates in our parameters. In addition, the modelling approach helps avoid multiplication of Type I errors by testing the effect of a given risk factor across both outcomes simultaneously. Despite these strengths, our finding should be interpreted with caution. We utilized data which is cross-sectional and could not establish causative effects. As with all studies, we could not account for all potential risk factors because the DHS data did not collect data on additional factors such as data on policies implemented in Ghana to address the problem of hypertension, including other individual, household and community level factors.

## Conclusion

The current study employed novel statistical methods, first of its kind in this area to capture residual similarity between SBP and DBP on women and their households, and residual household level variations in these outcomes, and identified critical risk factors. We addressed the statistical problem of ignoring correlation between SBP and DBP outcomes on the same woman and the household she lives which helps avoid imprecision in our model parameters, leading to better statistical inference and sound policy decisions. Substantial residual correlation between SBP and DBP on women and their households were found, while significant unobserved household level differences in SBP and DBP were observed. Women's age, obesity, education, toilet type and type of cooking fuel were found to be predictive of SBP and/or DBP. Our findings that substantial residual correlation between SBP and DBP on women and their households, and significant unobserved household level differences in SBP and DBP outcomes warrant further research to identify as-yet unidentified predictors of SBP and DBP outcomes in this population of women. There is also the need to conduct qualitative studies to establish why women from certain households are more likely to have increased SBP and DBP compared to women from other households and vice versa. We encourage researchers working on

hypertension predictors and the role of the higher-level hierarchy effects (e.g., household and community group levels) to jointly model SBP and DBP outcomes under multivariate response multilevel modelling framework as utilized in this study.

## Acknowledgments

Special thanks go to DHS program for providing access to the datasets used in the current study.

## Author Contributions

**Conceptualization:** Justice Moses K. Aheto.

**Data curation:** Justice Moses K. Aheto, Tracy Gates, Rahmatu Babah.

**Formal analysis:** Justice Moses K. Aheto.

**Investigation:** Justice Moses K. Aheto.

**Methodology:** Justice Moses K. Aheto.

**Project administration:** Justice Moses K. Aheto.

**Resources:** Justice Moses K. Aheto.

**Software:** Justice Moses K. Aheto.

**Supervision:** Justice Moses K. Aheto.

**Validation:** Justice Moses K. Aheto.

**Visualization:** Justice Moses K. Aheto.

**Writing – original draft:** Justice Moses K. Aheto, Tracy Gates, Rahmatu Babah, Wisdom Takramah.

**Writing – review & editing:** Justice Moses K. Aheto, Tracy Gates, Rahmatu Babah, Wisdom Takramah.

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
