## [Decision Letter · Decision Letter 0]

25 Jan 2023

PGPH-D-22-01851

Joint modelling of systolic and diastolic blood pressure and its associated factors among women in Ghana: Multivariate response multilevel modelling methods

Dear Dr. Aheto,

Thank you for submitting your manuscript to PLOS Global Public Health. After careful consideration, we feel that it has merit but does not fully meet PLOS Global Public Health’s publication criteria as it currently stands. Therefore, we invite you to submit a revised version of the manuscript that addresses the points raised during the review process.

We look forward to receiving your revised manuscript.

Kind regards,

Rajesh Sharma, Ph.D.

Academic Editor

Journal Requirements:

2. Please provide separate figure files in .tif or .eps format only and remove any figures embedded in your manuscript file. Please also ensure that all files are under our size limit of 10MB.

3. Figs 1-4: please (a) provide a direct link to the base layer of the map (i.e., the country or region border shape) and ensure this is also included in the figure legend; and (b) provide a link to the terms of use / license information for the base layer image or shapefile. We cannot publish proprietary or copyrighted maps (e.g. Google Maps, Mapquest) and the terms of use for your map base layer must be compatible with our CC-BY 4.0 license. 

Additional Editor Comments (if provided):

Reviewers' comments:

Reviewer's Responses to Questions

**Comments to the Author**

1. Does this manuscript meet PLOS Global Public Health’s publication criteria? Is the manuscript technically sound, and do the data support the conclusions? The manuscript must describe methodologically and ethically rigorous research with conclusions that are appropriately drawn based on the data presented.

Reviewer #1: Yes

Reviewer #2: Yes

2. Has the statistical analysis been performed appropriately and rigorously?

Reviewer #1: Yes

Reviewer #2: Yes

3. Have the authors made all data underlying the findings in their manuscript fully available (please refer to the Data Availability Statement at the start of the manuscript PDF file)?

Reviewer #1: Yes

Reviewer #2: Yes

4. Is the manuscript presented in an intelligible fashion and written in standard English?

Reviewer #1: No

Reviewer #2: Yes

5. Review Comments to the Author

Reviewer #1: there are a couple of areas that need editorial work. some paragraphs are too long, some sentences are a mouthful. just basic edits. This is especially so in the discussion and conclusions sections of the manuscript

Reviewer #2: Title of manuscript: Joint modelling of systolic and diastolic blood pressure and its associated factors among women in Ghana: Multivariate response multilevel modelling methods

The various factors affecting blood pressure in women of childbearing age is an important topic in public health as hypertension can lead to other cardiovascular diseases and mortality as well. Understanding the causative factors that lead to high blood pressure can help prevent many people from being hypertensive. The study was done on 4744 women residing in 3722 households using data from the 2014 Ghana Demographic and Health Survey. The authors recommend interventions to be targeted at high-risk groups including older generation and those who are obese.

Though the topic fits into the scope of the journal, the study has few weaknesses that needs attending to. The abstract section should be subtitled. There are some spelling and grammatical errors which needs correction. Some references are incomplete, such as reference number 3. Overall, the manuscript is well written.

Introduction

Page 1. Introduction. First paragraph

Hypertension is a significant risk factor for cardiovascular diseases, including stroke, myocardial infraction, heart failure, and atherosclerosis, which account for most of the non-communicable disease deaths globally. Change ‘infraction’ to ‘infarction’.

Page 2. Introduction. First paragraph

Understanding the influence of these demographic and social-ecological factors on hypertension is critical to addressing them, reducing prevalence, and improving treatment outcomes. Replace ‘social-ecological’ with ‘socio-ecological’.

Page 2. Introduction. Last paragraph

The multivariate response multilevel modelling is critical in this study because we are interested in modelling both SBP and SBP outcomes measured on same woman nested within a household and we need to draw conclusions about the degree to which the residual correlations depend on these women and their households. What about DBP?

Materials and methods

Statistical Analysis: Second paragraph

Insert ‘high’ blood pressure. The sentence should read as ‘Clinically, measurement of high blood pressure/hypertension is based on the combination of systolic blood pressure and diastolic blood pressure.’

Ethics Approval and Consent to Participate

An informed consent statement is read to the respondent, who may accept or decline to participate. Write this sentence in past tense.

6. PLOS authors have the option to publish the peer review history of their article (what does this mean?). If published, this will include your full peer review and any attached files.

**Do you want your identity to be public for this peer review?** For information about this choice, including consent withdrawal, please see our Privacy Policy.

Reviewer #1: **Yes: **Karani Magutah

Reviewer #2: No

---

## [Decision Letter · Decision Letter 1]

13 Mar 2023

PGPH-D-22-01851R1

Joint modelling of systolic and diastolic blood pressure and its associated factors among women in Ghana: Multivariate response multilevel modelling methods

Dear Dr. Aheto,

Thank you for submitting your manuscript to PLOS Global Public Health. After careful consideration, we feel that it has merit but does not fully meet PLOS Global Public Health’s publication criteria as it currently stands. Therefore, we invite you to submit a revised version of the manuscript that addresses the points raised during the review process.

We look forward to receiving your revised manuscript.

Kind regards,

Rajesh Sharma, Ph.D.

Academic Editor

Journal Requirements:

Additional Editor Comments (if provided):

Reviewers' comments:

Reviewer's Responses to Questions

**Comments to the Author**

1. If the authors have adequately addressed your comments raised in a previous round of review and you feel that this manuscript is now acceptable for publication, you may indicate that here to bypass the “Comments to the Author” section, enter your conflict of interest statement in the “Confidential to Editor” section, and submit your "Accept" recommendation.

Reviewer #1: (No Response)

2. Does this manuscript meet PLOS Global Public Health’s publication criteria? Is the manuscript technically sound, and do the data support the conclusions? The manuscript must describe methodologically and ethically rigorous research with conclusions that are appropriately drawn based on the data presented.

Reviewer #1: Yes

3. Has the statistical analysis been performed appropriately and rigorously?

Reviewer #1: Yes

4. Have the authors made all data underlying the findings in their manuscript fully available (please refer to the Data Availability Statement at the start of the manuscript PDF file)?

Reviewer #1: Yes

5. Is the manuscript presented in an intelligible fashion and written in standard English?

Reviewer #1: Yes

6. Review Comments to the Author

Reviewer #1: General remarks:

A number of issues raised earlier have been addressed and so left out in these new comments. Generally, the manuscripts looks much better now. A few areas here below may need address.

Abstract

1. I suggest dropping the less prevalent significant predictors of BP such as toilet type and cooking fuel from the abstract. They are curious and the abstract doesn’t provide for synthesis of how they affect BP. Maybe you only maintain the more prevalent predictors in abstract?

Introduction

1. Line 76. Authors say that when blood pressure is too high it exerts force on arterial walls thus leading to a critical reduction in blood flow. This is inaccurate. The worry in high blood pressure and hypertension is primarily destruction of blood vessels. If the author chooses to go that direction, they must discuss the physiologic mechanisms entailed.

Materials and methods

1. While there is probably no way out for the authors, data used is 8 years old and it is likely the described predictors have changed since. This is weakened further by the mention that such DHS data is available every 5 years. Was there any DHS report after 2014?

Results.

1. Line 272-275. The demographic information like mean age, numbers per age bracket, numbers with different levels of education, etc is lacking. This would go a long way in making readers appreciate the direction of the discussion being built.

2. Line 274. Paragraph one has a statement “However, a reduced figure of 6.9% of the women said they were ever told to have had hypertension” which is unclear.

3. Figure 3 needs some editorials (“observed” instead of “observe”)

Discussion

1. Line 404. The statement “The hypertension prevalence based on SBP, DBP, both SBP and DBP, and reported were 7.1%, 10.2%, 12.3%, and 6.9% respectively, with substantial regional differences in the hypertension prevalence for all the four measures” needs revision.

2. The discussion section has a lot of repeated results. Kindly revise this.

3. I was keen to see what discussion would come from the significant findings on toilet type and cooking fuel and how this fits in the larger scope on the topic but this is missing. For the other significant aspects as well, I wish likely mechanisms would have been discussed to make this work fit better in addressing a global public health problem from an understanding of the associated predictors.

Conclusion

1. This section is too long. It also has unnecessary details that would probably fit elsewhere (within discussion).

2. Some conclusions (like need to do screening and etc) are not drawn from the scope of the current study

7. PLOS authors have the option to publish the peer review history of their article (what does this mean?). If published, this will include your full peer review and any attached files.

**Do you want your identity to be public for this peer review?** For information about this choice, including consent withdrawal, please see our Privacy Policy.

Reviewer #1: **Yes: **KARANI MAGUTAH

---

## [Decision Letter · Decision Letter 2]

5 Apr 2023

Joint modelling of systolic and diastolic blood pressure and its associated factors among women in Ghana: Multivariate response multilevel modelling methods

PGPH-D-22-01851R2

Dear Dr. Aheto,

We are pleased to inform you that your manuscript 'Joint modelling of systolic and diastolic blood pressure and its associated factors among women in Ghana: Multivariate response multilevel modelling methods' has been provisionally accepted for publication in PLOS Global Public Health.

Best regards,

Rajesh Sharma, Ph.D.

Academic Editor

Reviewer Comments (if any, and for reference):

Reviewer's Responses to Questions

**Comments to the Author**

1. If the authors have adequately addressed your comments raised in a previous round of review and you feel that this manuscript is now acceptable for publication, you may indicate that here to bypass the “Comments to the Author” section, enter your conflict of interest statement in the “Confidential to Editor” section, and submit your "Accept" recommendation.

Reviewer #1: All comments have been addressed

2. Does this manuscript meet PLOS Global Public Health’s publication criteria? Is the manuscript technically sound, and do the data support the conclusions? The manuscript must describe methodologically and ethically rigorous research with conclusions that are appropriately drawn based on the data presented.

Reviewer #1: Yes

3. Has the statistical analysis been performed appropriately and rigorously?

Reviewer #1: Yes

4. Have the authors made all data underlying the findings in their manuscript fully available (please refer to the Data Availability Statement at the start of the manuscript PDF file)?

Reviewer #1: Yes

5. Is the manuscript presented in an intelligible fashion and written in standard English?

Reviewer #1: Yes

6. Review Comments to the Author

Reviewer #1: The manuscript flows well and i believe communicates a very important scientific message regards modelling of blood pressure variables. This is a great public health piece.

7. PLOS authors have the option to publish the peer review history of their article (what does this mean?). If published, this will include your full peer review and any attached files.

**Do you want your identity to be public for this peer review?** For information about this choice, including consent withdrawal, please see our Privacy Policy.

Reviewer #1: **Yes: **Karani Magutah
